# Holi-Spatial: Evolving Video Streams into Holistic 3D Spatial Intelligence

**Yuanyuan Gao**[1 2 *]  **Hao Li**[2 3 *]  **Yifei Liu**[1 4 *]  **Xinhao Ji**[1 5 *]  **Yuning Gong**[1 6 *]  **Yuanjun Liao**[6]  **Fangfu Liu**[7]
**Manyuan Zhang**[8]  **Yuchen Yang**[9]  **Dan Xu**[10]  **Xue Yang**[11]  **Huaxi Huang**[1]  **Hongjie Zhang**[1]  **Ziwei Liu**[3]
**Xiao Sun**[1]  **Dingwen Zhang**[2 †]  **Zhihang Zhong**[11 †]

*Equal Contribution; †Corresponding Authors

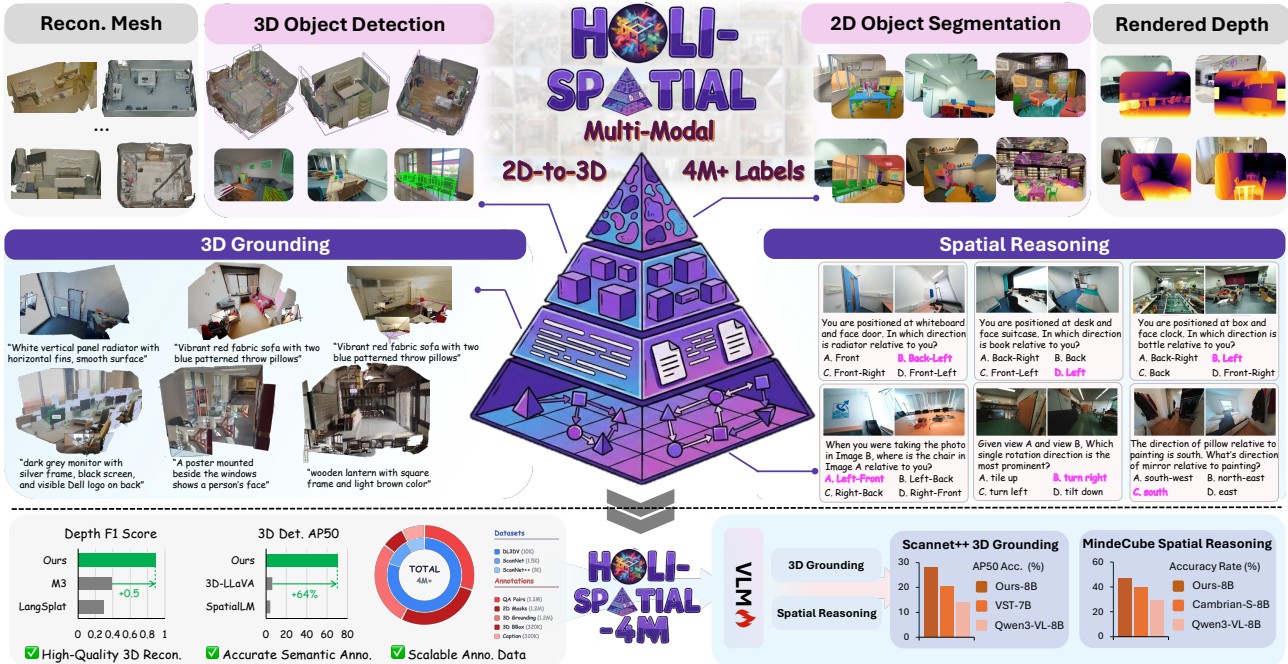

**Figure 1.** **We introduce Holi-Spatial, the first fully automated pipeline capable of converting raw video streams into holistic 3D spatial annotations without human intervention.** Compared to state-of-the-art methods, Holi-Spatial achieves a significant leap in annotation quality, improving multi-view depth estimation by 0.5 F1 and boosting 3D detection AP50 by a remarkable 64% on ScanNet (Dai et al., 2017). Based on this, we introduce **Holi-Spatial-4M**, a large-scale dataset that effectively empowers Vision-Language Models. As shown, fine-tuning Qwen3-VL on Holi-Spatial-4M leads to state-of-the-art performance, with a 15% AP50 gain on ScanNet++ (Yeshwanth et al., 2023) and a 7.9% accuracy rise on MMSI-Bench (Yang et al., 2025b). **Importantly, because the entire annotation pipeline is automatic, it can be further scaled up as resources permit.**

## Abstract

The pursuit of spatial intelligence fundamentally relies on access to large-scale, fine-grained 3D data. However, existing approaches predominantly construct spatial understanding benchmarks by generating question–answer (QA) pairs from a limited number of manually annotated datasets, rather than systematically annotating new large-scale 3D scenes from raw web data. As a result, their scalability is severely constrained, and model performance is further hindered by domain gaps inherent in these narrowly curated datasets. In this work, we propose **Holi-Spatial**, the first fully automated, large-scale, spatially-aware multimodal dataset, constructed from raw

[1]Shanghai AI Lab  [2]Northwestern Polytechnical University  [3]Nanyang Technological University  [4]Beihang University  [5]Peking University  [6]Sichuan University  [7]Tsinghua University  [8]The Chinese University of Hong Kong  [9]Fudan University  [10]Hong Kong University of Science and Technology  [11]Shanghai Jiao Tong University.  Correspondence to: Dingwen Zhang <zdw2006yyy@nwpu.edu.cn>, Zhihang Zhong <zhongzhihang@sjtu.edu.cn>.

*Proceedings of the 43rd International Conference on Machine Learning*, Seoul, South Korea. PMLR 306, 2026. Copyright 2026 by the author(s).

video inputs without human intervention, using the proposed data curation pipeline. Holi-Spatial supports multi-level spatial supervision, ranging from geometrically accurate 3D Gaussian Splatting (3DGS) reconstructions with rendered depth maps to object-level and relational semantic annotations, together with corresponding spatial Question–Answer (QA) pairs. Following a principled and systematic pipeline, we further construct **Holi-Spatial-4M**, the first large-scale, high-quality 3D semantic dataset, containing 12K optimized 3DGS scenes, 1.3M 2D masks, 320K 3D bounding boxes, 320K instance captions, 1.2M 3D grounding instances, and 1.2M spatial QA pairs spanning diverse geometric, relational, and semantic reasoning tasks. Holi-Spatial demonstrates exceptional performance in data curation quality, significantly outperforming existing feed-forward and per-scene optimized methods on datasets such as ScanNet, ScanNet++, and DL3DV. Furthermore, fine-tuning Vision-Language Models (VLMs) on spatial reasoning tasks using this dataset has also led to substantial improvements in model performance.

*Table 1.* **Overview of Pipeline Capabilities.** We compare input modalities and output tasks across different paradigms, ranging from 2D VLMs and 3D-VLMs to 3D-GS-based understanding methods. Holi-Spatial serves as a unified framework supporting diverse spatial tasks without relying on 3D priors.

| Method | Inputs | | Outputs | | | | |
|---|---|---|---|---|---|---|---|
| | Images | 3D | Depth | 2D Seg. | 3D Det. | Grounding | QA |
| *2D-VLM Methods* | | | | | | | |
| SAM3 | ✓ | ✗ | ✗ | ✓ | ✗ | ✗ | ✗ |
| SA2VA | ✓ | ✗ | ✗ | ✓ | ✗ | ✓ | ✓ |
| *3D-VLM Methods* | | | | | | | |
| SpatialLM | ✗ | ✓ | ✗ | ✗ | ✓ | ✓ | ✓ |
| LLaVA-3D | ✗ | ✓ | ✗ | ✗ | ✓ | ✓ | ✓ |
| SceneScript | ✗ | ✓ | ✗ | ✗ | ✓ | ✗ | ✗ |
| *3D-GS based Understanding Methods* | | | | | | | |
| M3-Spatial | ✓ | ✗ | ✓ | ✓ | ✗ | ✓ | ✗ |
| LangSplat | ✓ | ✗ | ✓ | ✓ | ✗ | ✗ | ✗ |
| **Holi-Spatial** | ✓ | ✓ | ✓ | ✓ | ✓ | ✓ | ✓ |

# 1. Introduction

Spatial intelligence (Feng et al., 2025b) is a fundamental bridge toward enabling large models to understand the real 3D world. It requires large multimodal models (LMMs) (Bai et al., 2025a; Team et al., 2023; Bai et al., 2025b) to move beyond 2D, language-centric perception and develop robust 3D spatial abilities to perceive, ground, and reason about the

3D world from visual inputs. Such capabilities hold great promise for a wide range of real-world applications, including robotic manipulation (Zhang et al., 2025b; Qu et al., 2025) and navigation (He et al., 2025), scene editing (Wang et al., 2025b), and augmented reality (Jiang et al., 2025).

However, a key limitation is the scarcity and imbalance of raw spatial data. Prior methods (Wu et al., 2025; Cai et al., 2025a; Yin et al., 2025; Yang et al., 2025a) typically curate spatial supervision by generating QA pairs from a small set of manually annotated 3D datasets (*e.g.*, ScanNet (Dai et al., 2017) and ScanNet++ (Yeshwanth et al., 2023)) or by naively applying feed-forward perception models (Carion et al., 2025) to single-image data (Gupta et al., 2019). While these strategies improve over general VLMs (Bai et al., 2025a;b; Team et al., 2023), they are difficult to scale due to reliance on specialized scanning hardware and human-in-the-loop annotation, and they often provide limited semantic coverage (*e.g.*, only 50 labeled classes in ScanNet (Dai et al., 2017)).

To address these limitations, we note that recent advances in relevant AI tools (Carion et al., 2025; Team et al., 2023; Wang et al., 2025a; Lin et al., 2025) have exceeded expectations; by systematically composing them, we may build an automated spatial annotation engine that can even outperform human annotations, enabling a positive data flywheel. Thus, we present **Holi-Spatial**, a fully automated framework that converts raw video streams into high-fidelity 3D geometry together with holistic semantic annotations *without* requiring any explicit 3D sensors or human-in-the-loop labeling. As summarized in Table 1, Holi-Spatial unifies a broad set of spatial tasks, including 3D reconstruction, novel view synthesis (NVS), depth rendering, 2D instance segmentation, instance captioning, 3D bounding boxes, 3D grounding, and spatial QA. Holi-Spatial is composed of three stages. (i) *Geometric Optimization:* we initialize from monocular priors (Depth-Anything-V3 (Lin et al., 2025)) and optimize a 3D Gaussian Splatting (3DGS) scene under geometric supervision like (Chen et al., 2024; Gao et al., 2025b) to sharpen structure and suppress floaters. (ii) *Image-level Perception:* We sample keyframes, use a VLM to infer

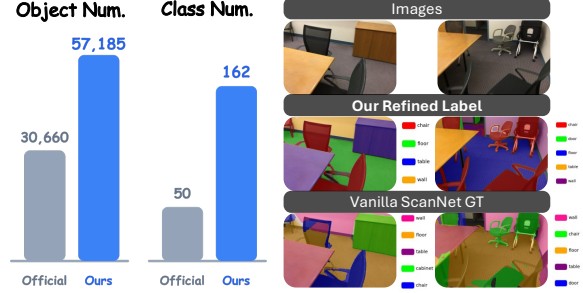

*Figure 2.* **Comparison of our refined annotations with the official annotations on the ScanNet dataset (Dai et al., 2017).**

open-vocabulary categories, and, motivated by the rapid progress of segmentation models (Ravi et al., 2025; Ren et al., 2024; Zhang et al., 2025a; Kirillov et al., 2023), we adopt the state-of-the-art SAM3 (Carion et al., 2025) to produce high-quality open-set masks for each image. (iii) *Scene-level Lift and Refinement:* We lift the 2D masks into 3D by back-projecting their pixels using rendered depth and camera intrinsics, and transforming the resulting points into the world frame using recovered camera poses. The resulting 3D points serve as instance candidates. We merge redundant candidates across views by checking their bounding box IoU, and use a VLM-based agent to filter low-confidence candidates; for the merged and most reliable instances, we generate detailed captions and further construct grounding and QA pairs for training VLM's 3D Grounding and Spatial Reasoning ability.

We evaluate the proposed curation pipeline on established benchmarks, including ScanNet (Dai et al., 2017), ScanNet++ (Yeshwanth et al., 2023), and DL3DV-10K (Ling et al., 2024), where we create open-vocabulary GT by additionally re-annotating the scenes with human supervision to enable reliable evaluation of fine-grained objects. As shown in Figure 1, Holi-Spatial consistently outperforms both model-based baselines and 3DGS-based optimization approaches; on ScanNet++ (Yeshwanth et al., 2023), we improve multi-view depth estimation by up to 0.5 F1 and boost 3D detection $AP_{50}$ by 64%.

Leveraging this pipeline, we construct and release **Holi-Spatial-4M** from diverse video sources (ScanNet (Dai et al., 2017), ScanNet++ (Yeshwanth et al., 2023), and DL3DV-10K (Ling et al., 2024)). The dataset contains 12K optimized 3DGS scenes with multi-view consistent depth renderings, as well as 1.2M 2D masks, 1.2M 3D Grounding, 320K 3D bounding boxes, 320K instance captions, and 1.3M spatial QA pairs. Beyond scale, our annotations exhibit higher granularity and improved boundaries compared to official labels on Scannet (Dai et al., 2017) ( Figure 2). To verify that Holi-Spatial-4M improves spatial intelligence, we fine-tune the Qwen3-VL family (Bai et al., 2025a;b) on our curated data. This yields consistent gains on public benchmarks, including a 15% $AP_{50}$ improvement on ScanNet++ (Yeshwanth et al., 2023) for 3D grounding and a 7.9% accuracy increase on MMSI-Bench (Yang et al., 2025b).

## 2. Related Work

### 2.1. Data Scalability in Spatial Intelligence

Recent advancements in Large Multimodal Models (LMMs) have demonstrated remarkable proficiency in 2D visual understanding and planar reasoning (Team et al., 2023; Bai et al., 2025a;b), but still exhibit a significant lag behind human capabilities in Spatial Intelligence (Cai et al., 2025b;

Yang et al., 2025b; Yin et al., 2025). We attribute this gap primarily to a critical disparity in **raw data diversity**. Unlike 2D datasets such as LAION-5B (Schuhmann et al., 2022) provide *billion-scale* unique image inputs, the spatial intelligence domain suffers from severe scene scarcity. Although recent spatial datasets such as SenseNova-SI-800K (Cai et al., 2025a) and VST-4M (Yang et al., 2025a) boast million-level annotations, these are predominantly derived from a tiny pool of only *a few thousand* static 3D scans (*e.g.*, ScanNet). This reliance on a narrow set of environments fundamentally limits generalization, motivating us to construct a more scalable data curation framework. Instead of relying on a limited set of labeled 3D assets, we aim to unlock the potential of abundant web videos, automatically generating dense spatial annotations to fuel the next generation of spatial intelligence.

### 2.2. Methods for Spatial Intelligence

Efforts to enhance spatial intelligence in multimodal models primarily follow three approaches: (1) *3D-native LMMs*, a line of works (Deng et al., 2025; Zhu et al., 2024; Wu et al., 2025; Mao et al., 2025; Zheng et al., 2025) directly consumes explicit 3D observations (*e.g.*, point clouds, meshes, or multi-view RGB-D) and performs reasoning in 3D space, such as SpatialLM (Mao et al., 2025) and LLaVA-3D (Zhu et al., 2024). (2) *2D-Centric spatial LMMs* (Yang et al., 2025a; Cai et al., 2025a; Yang et al., 2025c; Yin et al., 2025) improve performance primarily by scaling up training datasets and training recipes on spatial perception and reasoning. For example, VST (Yang et al., 2025a) adopts 4.1M samples for SFT and RL, while Cambrian-S (Yang et al., 2025c) develops the VSI-590K dataset to enhance spatial video understanding. Orthogonal to dataset-centric scaling for spatial reasoning, efficient adaptation of vision backbones has also been explored; for example, EA-ViT introduces an elastic adaptation strategy for Vision Transformers to improve adaptation efficiency across downstream tasks (Zhu et al., 2025a). (3) *3DGS-based methods* (Qin et al., 2024; Zou et al., 2025; Li et al., 2024; Gao et al., 2026) leverage 3DGS (Kerbl et al., 2023) as an explicit scene representation and optimize it to align geometry with language / vision signals (Radford et al., 2021). For example, M3-Spatial (Zou et al., 2025) augments a per-scene 3DGS reconstruction with language-aligned features to support open-vocabulary 3D grounding within the optimized scene.

Despite notable progress, existing paradigms face fundamental scalability bottlenecks. Approaches (1) and (2) rely heavily on human-annotated 3D data or manually curated scans, making dataset expansion costly and intrinsically limited by 3D acquisition and annotation overhead. Approach (3), based on optimization-driven 3DGS, requires per-scene training or finetuning, which is time-consuming and often

unstable, hindering large-scale deployment. In contrast, Holi-Spatial reframes spatial data curation as a scalable, non-human pipeline that converts raw videos into *annotated high-fidelity 3D scenes*, enabling automatic generation of large-scale spatial-understanding data.

## 3. Method

Here we introduce the core steps of our curation framework, as illustrated in Figure 3. It is composed of three stages: (i) **Geometric Optimization** distills high-fidelity 3D structure from raw videos; (ii) **Image-level Perception** extracts spatially consistent object label from VLM and produces high-quality 2D masks on keyframes; and (iii) **Scene-level Refinement** lifts, merges, filters, verifies, and captions instances in 3D to generate dense, reliable spatial annotations.

**Geometric Optimization.** The primary objective of this stage is to distill raw video streams into high-fidelity geometric structures, serving as a robust prerequisite for precise spatial annotations. To achieve this, we first employ Structure-from-Motion (Schönberger & Frahm, 2016) to resolve accurate camera intrinsics and extrinsics, followed by leveraging a state-of-the-art spatial foundation model (Lin et al., 2025) to initialize a dense, structurally coherent point cloud. However, feed-forward depth estimations inevitably contain noise and outliers. To address this, we incorporate 3D Gaussian Splatting (3DGS) (Kerbl et al., 2023) for per-scene optimization. Specifically, we integrate geometric regularization inspired by the surface reconstruction GS method (Chen et al., 2024; Li et al., 2024; Gao et al., 2025b) to enforce multi-view depth consistency. This process effectively eliminates floaters that would otherwise interfere with 3D bounding box generation, yielding a clean and consistent scene representation aligned with physical surfaces.

**Image-level Perception.** We uniformly sample a set of keyframes $\mathcal{I} = \{I_1, \ldots, I_T\}$ from the raw video stream. For each frame $I_t$, we employ Gemini3-Pro (Team et al., 2023) to generate a caption sequentially. Crucially, we maintain a dynamic class-label memory $\mathcal{M}_t$ to ensure semantic consistency. This memory accumulates recognized categories from previous frames ($I_{1:t-1}$) and instructs the VLM to prioritize reusing existing labels, formally updated as $\mathcal{M}_t = \mathcal{M}_{t-1} \cup \text{Extract}(I_t)$. Guided by the prompt derived from $\mathcal{M}_t$, SAM3 (Carion et al., 2025) performs open-vocabulary instance segmentation, producing a set of predictions $\mathcal{O}_t = \{(M_k, s_k)\}_{k=1}^N$, where $M_k$ represents the binary mask and $s_k$ denotes the confidence score.

Leveraging the refined depth map $D_t$ rendered from our optimized 3DGS, we unproject each pixel $\mathbf{u} = (u, v)$ in mask $M_k$ into a 3D point $\mathbf{P} \in \mathbb{R}^3$ via: $\mathbf{P} = D_t(\mathbf{u}) \cdot \mathbf{K}^{-1}\tilde{\mathbf{u}}$, where $\mathbf{K}$ is the camera intrinsic matrix and $\tilde{\mathbf{u}} = [u, v, 1]^\top$ is the homogeneous coordinate. However, directly computing

an Oriented Bounding Box (OBB) from these unprojected points often yields inaccurate bounds due to depth edge floaters around objects. These edge artifacts mainly arise from two sources: (i) **2D-level errors**, where SAM3 segmentation introduces boundary misalignment near object contours; and (ii) **3D-level noise**, where depth discontinuities naturally lead to unstable measurements and outliers. To suppress these floaters and improve grounding box precision, we seamlessly integrate a geometry-aware filtering strategy during the 2D-to-3D lifting process, as detailed in Figure 4. Specifically, this strategy mitigates both the 2D boundary errors via mask erosion and the 3D outliers through mesh-guided depth filtering, ensuring that the estimated initial 3D OBBs are derived from a highly reliable geometry subset.

**Scene-level Refinement.** The core motivation of this stage is a coarse-to-fine refinement strategy designed to distill high-fidelity annotations from noisy initial proposals, which is divided into three core parts:

1) Multi-View Merge & Post-Process. We first perform spatial clustering to consolidate redundant detections. Specifically, we iterate over all pairs of instances $p_i, p_j \in \mathcal{P}_{\text{init}}$ and merge them if they share the same category and have sufficient 3D overlap:

$$c_i = c_j \ \wedge \ \text{IoU}_{3D}(B_i, B_j) > \tau_{\text{merge}}, \qquad (1)$$

where $\tau_{\text{merge}}$ is set to $0.2$. It mitigates object fragmentation by consolidating disjoint observations of the same object (*e.g.*, parts of a large sofa) into a single 3D bounding box. For each merged instance, we update its attributes to preserve the most reliable observation: the confidence score is set to $s_k = \max(s_i, s_j)$, and we retain the source image index associated with this maximum. This selection mechanism grounds subsequent VLM captioning in the most informative visual perspective.

2) Confidence-Based Filtering & Refinement. We then refine the consolidated set $\mathcal{P}_{\text{merged}}$ based on the updated scores $s_k$. A tri-level decision rule is applied to each instance $p_k$:

$$\text{Action}(p_k) = \begin{cases} \text{keep}, & s_k \geq \tau_{\text{high}}, \\ \text{discard}, & s_k < \tau_{\text{low}}, \\ \text{verify}, & \tau_{\text{low}} \leq s_k < \tau_{\text{high}}, \end{cases} \qquad (2)$$

where thresholds are set to $\tau_{\text{high}} = 0.9$ and $\tau_{\text{low}} = 0.8$. For proposals in the *verify* band, we invoke a VLM-based agent equipped with an image zoom-in tool and a SAM3 re-segmentation tool to reassess the instance, yielding an updated confidence score $s'_k$. We keep the proposal if $s'_k \geq \tau_{\text{high}}$; otherwise, we discard it.

3) Upon establishing the final set of validated instances $\mathcal{P}_{\text{final}}$, we proceed to generate dense semantic annotations. For each instance $p_k \in \mathcal{P}_{\text{final}}$, we retrieve the optimal source

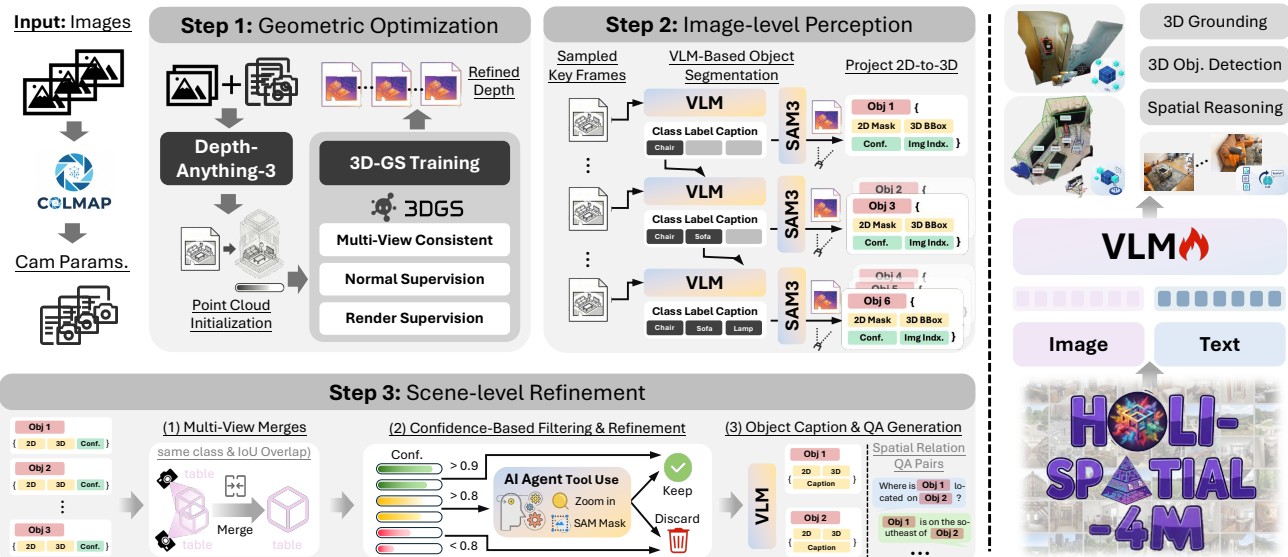

Figure 3. **Overview of the Holi-Spatial data curation pipeline.** The framework operates in three progressive stages: (1) **Geometric Optimization** distills high-fidelity 3D structure from video streams using 3DGS; (2) **Image-level Perception** lifts 2D VLM and SAM3 predictions into initial 3D proposals; and (3) **Scene-level Refinement** employs a coarse-to-fine strategy to merge, verify, and caption instances, yielding dense, high-quality spatial annotations. Finally, leveraging the generated **Holi-Spatial-4M** dataset, we directly fine-tune the Qwen-VL family for downstream tasks (*e.g.*, 3D grounding and spatial reasoning).

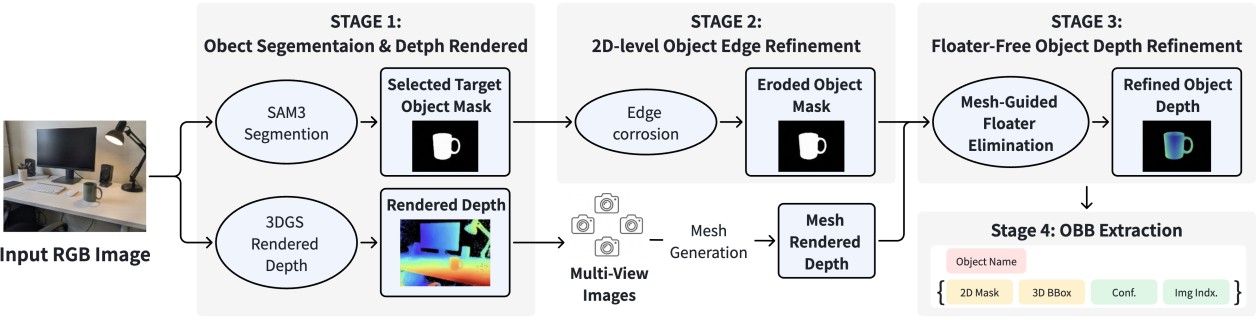

Figure 4. **Pipeline of 2D-to-3D OBB Generation.** We transform 2D object masks into initial 3D OBBs via depth projection, utilizing a four-step strategy to mitigate the impact of depth floaters. (1) We obtain an initial object depth map by combining 3DGS rendering with SAM3 instance segmentation. (2) To mitigate 2D boundary errors from SAM3, we erode the object mask near its contour and keep only the reliable interior region. (3) To remove 3D outliers caused by depth discontinuities, we use a multi-view-consistent mesh depth as guidance and filter inconsistent pixels in the 3DGS depth. (4) Finally, we estimate the initial 3D OBB from the refined point cloud, while preserving the associated 2D mask, confidence score, and source image index.

image $I_k^*$ corresponding to the retained high-confidence index from Step 1. Leveraging this most informative visual perspective, we employ Qwen3-VL-30B to generate a fine-grained caption for the object and procedurally synthesize a comprehensive suite of spatial QA pairs based on predefined templates, covering diverse tasks such as 3D grounding, spatial reasoning, and attribute identification.

## 4. Dataset

We introduce **Holi-Spatial-4M**, the first large-scale, multi-spatial-modal dataset constructed using the proposed automated curation pipeline. As illustrated in Figure 5, this dataset represents a significant advancement in scale, granularity, and task diversity compared to existing datasets.

**Data Composition and Scale.** Holi-Spatial-4M is derived from a diverse collection of raw video streams sourced from ScanNet (Dai et al., 2017), ScanNet++ (Yeshwanth et al., 2023), and DL3DV-10K (Ling et al., 2024). By processing these streams through our geometric optimization and scene-level refinement stages, we have curated over 12,000 optimized 3DGS scenes. As shown in Figure 5 (2), the dataset encompasses a total of 4 million+ high-quality spatial annotations. This includes 1.3M 2D instance masks, 1.2M 3D grounding pairs, 320K 3D bounding boxes, and 320K detailed instance captions, significantly surpassing

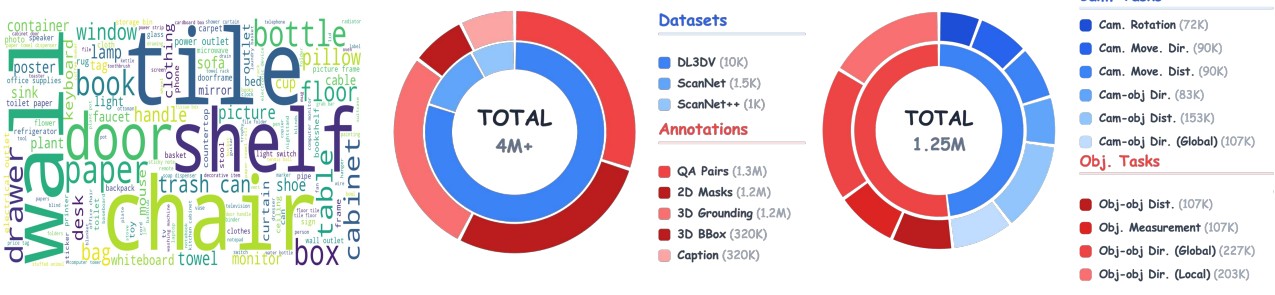

(1) Word Cloud of Different Object Classes  (2) Distribution of the Whole Dataset  (3) Categories Distribution of Spatial QA Pairs

*Figure 5.* **Comprehensive Statistics of Holi-Spatial-4M.** (1) **Object Diversity:** Word cloud showing the long-tailed distribution of open-vocabulary categories. (2) **Dataset Composition:** The inner ring displays source scenes (ScanNet (Dai et al., 2017), Scan-Net++ (Yeshwanth et al., 2023), DL3DV (Ling et al., 2024)), while the outer ring details the breakdown of over 4M generated spatial annotations. (3) **Spatial QA Taxonomy:** Distribution of 1.25M spatial QA pairs, categorized into Camera-centric tasks (*e.g.*, rotation, movement) and Object-centric tasks (*e.g.*, distance, direction).

*Table 2.* **Quantitative results of 3D spatial understanding on ScanNet, ScanNet++ and DL3DV datasets.** We group methods by their input modality and task definition. **3D Det**: 3D Object Detection (AP@25, AP@50); **2D Seg**: 2D Object Segmentation (IoU); **Depth**: Depth Estimation (F1-score). **Bold** indicates best results. ↑: higher is better. '—' denotes that the corresponding metric is not available.

| Method | ScanNet | | | | ScanNet++ | | | | DL3DV | | | |
|---|---|---|---|---|---|---|---|---|---|---|---|---|
| | Depth (↑) | 2D Seg. (↑) | 3D Det. (↑) | | Depth (↑) | 2D Seg. (↑) | 3D Det. (↑) | | Depth (↑) | 2D Seg. (↑) | 3D Det. (↑) | |
| | F1 | IoU | $AP_{25}$ | $AP_{50}$ | F1 | IoU | $AP_{25}$ | $AP_{50}$ | F1 | IoU | $AP_{25}$ | $AP_{50}$ |
| *2D-VLM Methods* | | | | | | | | | | | | |
| SAM3 | — | 0.63 | — | — | — | 0.50 | — | — | — | 0.66 | — | — |
| SA2VA | — | 0.64 | — | — | — | 0.25 | — | — | — | 0.44 | — | — |
| *3D-VLM Methods* | | | | | | | | | | | | |
| SpatialLM | — | — | 11.42 | 8.19 | — | — | 9.11 | 6.23 | — | — | 7.05 | 4.38 |
| LLaVA-3D | — | — | 9.13 | 6.86 | — | — | 12.2 | 4.80 | — | — | 6.83 | 4.11 |
| SceneScript | — | — | 8.97 | 3.54 | — | — | 9.86 | 4.42 | — | — | 5.65 | 3.98 |
| *3DGS-based Methods* | | | | | | | | | | | | |
| M3-Spatial | 0.32 | 0.22 | — | — | 0.39 | 0.11 | — | — | 0.23 | 0.13 | — | — |
| LangSplat | 0.19 | 0.36 | — | — | 0.21 | 0.06 | — | — | 0.18 | 0.24 | — | — |
| **Holi-Spatial (Ours)** | **0.98** | **0.66** | **76.60** | **67.00** | **0.89** | **0.64** | **81.06** | **70.05** | **0.78** | **0.71** | **62.89** | **52.67** |

the scale of manual annotations in the original datasets.

**Open-Vocabulary Diversity.** Unlike traditional datasets limited to a closed set of categories, Holi-Spatial-4M leverages the open-world knowledge of VLMs to annotate a vast array of objects. Figure 5 (1) presents a word cloud of the object categories, highlighting the dataset's coverage of fine-grained indoor items. This semantic richness is crucial for training models capable of generalized spatial understanding in real-world environments.

**Spatial Question-Answering Pairs.** To empower Large Multimodal Language Models with robust spatial reasoning capabilities, we generate 1.25M Spatial QA pairs, structured into a comprehensive taxonomy. As detailed in Figure 5 (3), these QAs are divided into two primary categories: 1) **Camera-centric Tasks** (blue sector), which challenge the model to understand ego-centric spatial changes such as *Camera Rotation* and *Movement Direction*; 2) **Object-centric Tasks** (red sector), which focus on allocentric reasoning, including *Object-to-Object Distance*, *Global/Local Direction*, and *Size Measurement*. This balanced distribu-

tion ensures that models trained on Holi-Spatial-4M develop a holistic understanding of 3D space.

## 5. Experiment

### 5.1. Framework Evaluation

**Settings.** To conduct a fair comparison of our framework, we randomly sampled 10 scenes from each of the ScanNet, ScanNet++, and DL3DV-10K datasets. For these selected scenes, we manually annotated 2D instance masks and 3D bounding boxes to serve as the evaluation ground truth. For depth evaluation, we directly utilized the official ground truth depth maps provided by the respective scenes.

**Results.** Table 2 highlights Holi-Spatial as the sole framework capable of simultaneously generating high-quality predictions across 3D object detection, 2D segmentation, and depth estimation, whereas prior works typically specialize in single modalities. First, in terms of geometric fidelity, our method significantly outperforms 3DGS-based baselines with a Depth F1-score of 0.89 on ScanNet++ (Yeshwanth

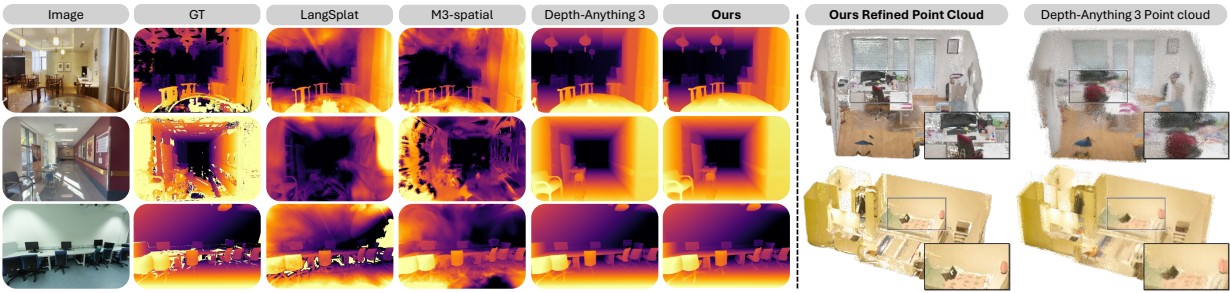

*Figure 6.* **Qualitative comparison of multi-view depth on ScanNet++.** We visualize depth maps from GT and baselines (LangSplat, M3-Spatial, Depth-Anything-V3) versus Holi-Spatial. Right: point clouds obtained by multi-view back-projection, where Holi-Spatial produces cleaner geometry with substantially fewer ghosting artifacts and floaters.

et al., 2023) compared to 0.39 for M3-Spatial (Zou et al., 2025). As shown in Figure 6, the quantitative depth visualization indicates that both DA3 and our method already provide strong relative depth cues compared with other baselines; to further highlight our multi-view advantage, the right panel visualizes the point cloud obtained by multi-view projection, where our results exhibit almost no ghosting and significantly fewer floaters.

Regarding 2D segmentation, our framework demonstrates superior quality, reaching 0.64 IoU compared to 0.25 for SA2VA (Yuan et al., 2025). As illustrated in the second row of Figure 7, SAM3 fails to segment the distant mirror, whereas our method, by leveraging multi-view information, successfully segments such challenging and unclear instances. This improvement is attributed to the synergy between geometric and semantic priors: while the VLM agent provides robust semantic reasoning at the image level, our explicit integration of multi-view information compensates for incomplete observations from single images.

Regarding 3D object detection, building on the high-quality geometry and refined semantics above, Holi-Spatial achieves dominant performance in 3D object detection. On ScanNet++, we report an $AP_{25}$ of 81.06, exceeding the state-of-the-art 3D-VLM (LLaVA-3D, 12.2 $AP_{25}$) by an order of magnitude. Furthermore, Figure 8 provides a quantitative comparison of the predicted instances, where Holi-Spatial recovers substantially more objects with accurate labels and tightly aligned 3D bounding boxes, demonstrating strong semantic fidelity and geometric precision.

### 5.2. VLM Finetuning Evaluation

**Settings**. For the spatial reasoning task, we finetune Qwen3-VL (Bai et al., 2025a) families using the 1.2M spatial QA pairs in our **Holi-Spatial-4M** dataset for 1 epoch with batch size 1024, and evaluate its performance on MMSI-Bench (Yang et al., 2025b) and MindCube (Yin et al., 2025). For 3D grounding task, we finetune Qwen3-VL-8B (Bai et al., 2025a) using 1.2M 3D grounding pairs in our Holi-Spatial-4M dataset. All models are trained for a single

epoch with a total batch size of 1024. Training is conducted on 32 NVIDIA H800 GPUs (80GB).

**Spatial Reasoning**. Table 3 shows the QA training results. Our model consistently improves both 2B and 8B models after finetuning, improving model's spatial understanding thanks to our high quality 3D curated data and QA pairs. More QA examples are provided in the Appendix.

**3D Grounding**. The results are summarized in Table 4. In particular, our method achieves an $AP_{50}$ of 27.98, exceeding the strongest baseline by 14.48 AP points, which we attribute to fine-tuning on our curated dataset with stronger 3D grounding supervision. As illustrated in Figure 10, baseline models such as Qwen3-VL (Bai et al., 2025a), trained primarily on single-view or anchor-view data, exhibit a clear viewpoint bias and fail to reliably ground objects across different views or at varying spatial depths.

### 5.3. Ablation

**Geometric Training**. In this section, we conduct ablation experiments on geometric training. As shown in Figure 9, we compare results obtained using the original DA3 depth (ID. 1) and those using our GS-refined depth (ID. 2) in the ScanNet++ scene. The visualization indicates that directly using DA3 (Lin et al., 2025) depth often introduces ghosting artifacts when projecting from multiple views. For example, in the shelf1curtain case, such ghosting leads to erroneous clustering and incorrectly splits one instance into multiple instances, which is also clearly observable in the 3D viewer at the bottom right. In contrast, our refined depth is more multi-view consistent, resulting in more reliable clustering and cleaner instance reconstruction. Moreover, as shown in Table 5, the GS-refined depth substantially outperforms direct use of DA3 (Lin et al., 2025) depth.

**Confidence Filter**. As shown in Table 5, we conduct ablation studies on the SAM3 (Carion et al., 2025) confidence filter mechanism in the ScanNet++ scene. The confidence filter improves precision (from 0.35 to 0.67) by suppressing false positives: as illustrated in Figure 9 (ID. 3/ID. 4), SAM3 may assign an incorrect category (*e.g.*, classifying

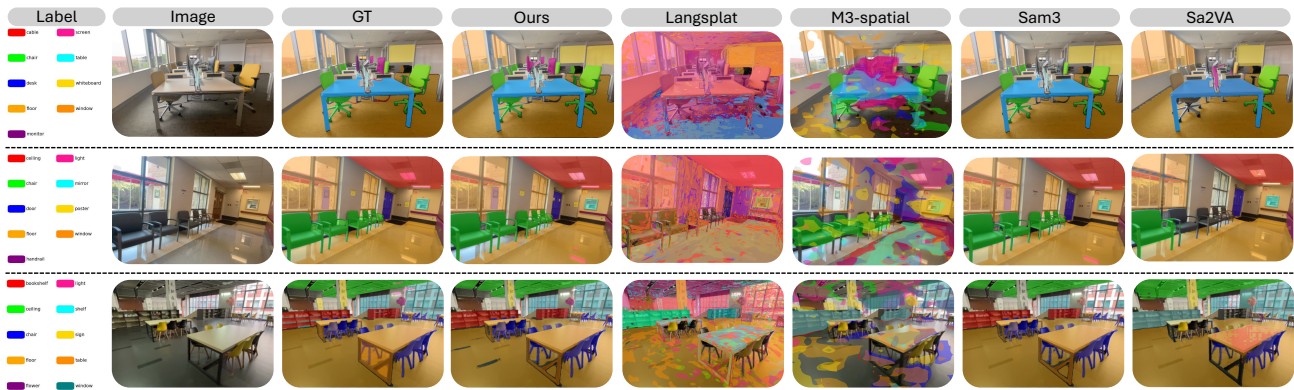

*Figure 7.* **Qualitative comparison of open-vocabulary 2D instance segmentation on the same scenes.** Holi-Spatial yields sharper boundaries, more complete masks under occlusion, and more accurate fine-grained categories.

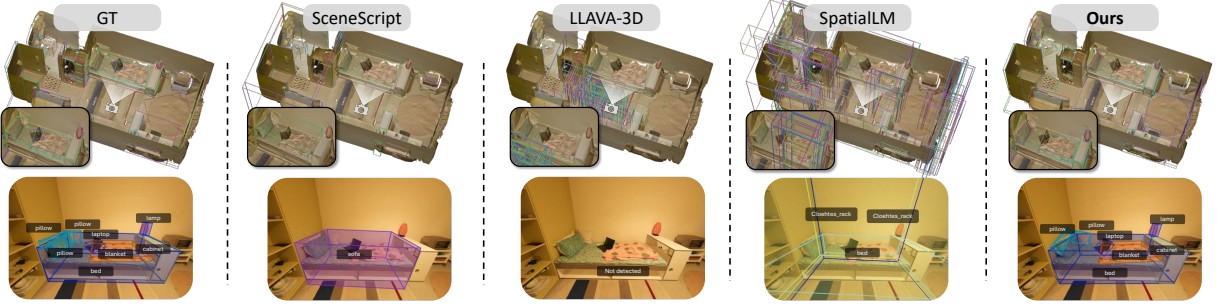

*Figure 8.* **Qualitative comparison of 3D object detection on ScanNet++.** We visualize predicted 3D bounding boxes from SceneScript, LLaVA-3D, SpatialLM, and Holi-Spatial. Holi-Spatial produces tighter boxes and more correct categories.

*Table 4.* **Quantitative results of 3D Grounding on Scan-Net++ (Yeshwanth et al., 2023) dataset.**

| Method | $AP_{15}$ | $AP_{25}$ | $AP_{50}$ |
|---|---|---|---|
| VST-7B-SFT | 17.29 | 14.50 | 11.20 |
| Qwen3-VL-8B | 19.82 | 16.80 | 13.50 |
| Qwen3-VL-8B + Ours | **35.52** | **31.94** | **27.98** |

multiple machines as *vending machine*), and confidence-based filtering helps remove such misclassified predictions.

**Agent Refinement**. According to our tests, as shown in Table 5, Confidence filtering also reduces recall (from 0.74 to 0.69) when comparing ID. 3 and ID. 4, since it tends to discard visually challenging yet correct instances. For example, in Figure 9, ID. 4, it shows objects such as a hair dryer among clutter and a cart heavily occluded by buckets and cleaning tools can receive low confidence and thus be mistakenly filtered out. To address this trade-off, we introduce an agent-based (VLM) verification step to reconsider borderline cases instead of discarding them directly, which

*Table 3.* **Quantitative results of different models (Ouyang et al., 2025; Zhu et al., 2025b; Wu et al., 2025) on MMSI-bench (Yang et al., 2025b), MindCube (Yin et al., 2025), ViewSpatial (Li et al., 2025a), and SparBench-tiny (Zhang et al., 2026) benchmarks.** Bold indicates the best performance.

| Model | MMSI-Bench | MindCube | ViewSpatial | SparBench-tiny |
|---|---|---|---|---|
| VST-SFT-3B | 30.2 | 35.9 | 52.89 | 37.71 |
| Cambrian-S-3B | 25.2 | 32.5 | 40.97 | 33.02 |
| VST-SFT-7B | 32.0 | 39.7 | 50.53 | 46.61 |
| Cambrian-S-7B | 25.8 | 39.6 | 41.28 | 37.91 |
| SpaceR-SFT-7B | 27.4 | 37.9 | — | — |
| Intern3-VL-8B | 28.0 | 41.5 | 38.66 | 35.86 |
| Spatial-MLLM | 27.0 | 32.1 | 34.66 | 35.31 |
| Qwen3-VL-2B | 26.1 | 33.5 | — | — |
| Qwen3-VL-2B + Ours | **27.6** | **44.0** | — | — |
| Qwen3-VL-8B | 31.1 | 29.4 | 42.35 | 39.83 |
| Qwen3-VL-8B + Ours | **32.6** | **49.1** | **50.12** | **46.28** |

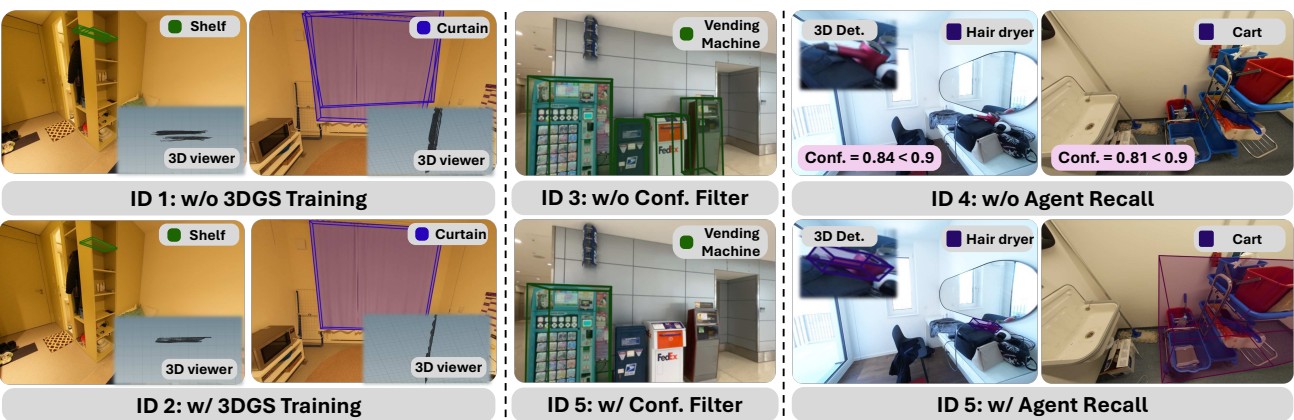

Figure 9. **Stage-wise visualization of scene-level refinement.** Detailed discussion is include in Section 5.3

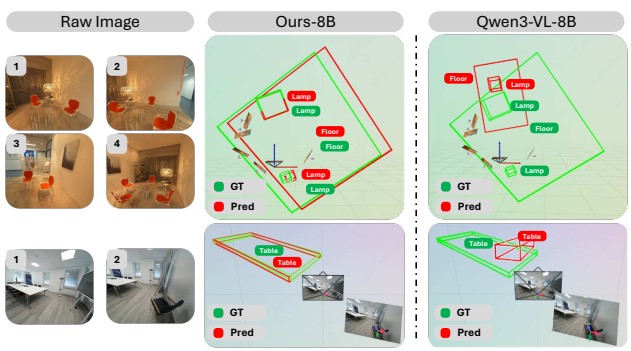

Figure 10. Given the same input images and queried objects, our predicted boxes align better with ground truth, indicating improved spatial localization.

Table 5. **Ablation study on depth refinement, confidence filtering, and agent recall.** ID. 1–2 (Step 1) compare depth sources, while ID. 3–5 (Step 3) evaluate post-processing modules. $P_{25}$ and $R_{25}$ denote precision and recall under IoU 25%.

| ID | DA3 Depth | 3DGS Training | Conf. Filter | Agent Recall | $P_{25}$ | $R_{25}$ |
|---|---|---|---|---|---|---|
| *Step 1: Geometric Optimization* | | | | | | |
| 1 | ✓ | ✗ | ✓ | ✓ | 0.13 | 0.31 |
| 2 | ✓ | ✓ | ✓ | ✓ | 0.81 | 0.89 |
| *Step 3: Scene-Level Refinement* | | | | | | |
| 3 | ✓ | ✓ | ✗ | ✗ | 0.35 | 0.74 |
| 4 | ✓ | ✓ | ✓ | ✗ | 0.67 | 0.69 |
| 5 | ✓ | ✓ | ✓ | ✓ | 0.81 | 0.89 |

recovers true positives with low confidence. Overall, the confidence filter and the agent recall complement each other and together yield the best precision–recall balance.

# 6. Conclusion

In this paper, we present **Holi-Spatial**, a fully automated pipeline that converts raw videos into high-fidelity 3D geometry and holistic spatial annotations. By combining 3DGS-based geometric optimization, open-vocabulary perception, and scene-level lifting/refinement, it produces multi-level supervision (rendered depth, 2D masks, 3D boxes, instance captions, and spatial QA). We further release **Holi-Spatial-4M** with 12K optimized 3DGS scenes, 1.3M masks, 320K 3D boxes/captions, 1.2M 3D grounding instances, and 1.2M QA pairs, and show on ScanNet/ScanNet++/DL3DV that fine-tuning VLMs on Holi-Spatial-4M consistently improves 3D grounding and spatial reasoning.

# Acknowledgment

This research was supported by the Shanghai AI Laboratory and the National Natural Science Foundation of China under Grants 62293543 and 62322605.

# Impact Statement

This work introduces Holi-Spatial, a fully automated data curation pipeline that converts raw video streams into high-fidelity 3D geometry together with holistic spatial annotations, enabling the construction of large-scale spatially-aware multimodal datasets.

However, Holi-Spatial still has limitations. The pipeline relies on multiple upstream components and per-scene optimization, which can be computationally expensive and may degrade under challenging videos (e.g., limited viewpoints, motion blur, heavy occlusion, or dynamic objects). A promising future direction is to extend our framework to more diverse real-world scenarios by incorporating recent advances in dynamic scene reconstruction (Wu et al., 2024; Luiten et al., 2024; Li et al., 2025b; Yan et al., 2024). Moreover, Open-vocabulary semantic labeling may also inherit biases or errors from foundation models, making robust verification and uncertainty estimation important future directions. We plan to improve efficiency (e.g., adaptive early stopping and better confidence-based validation and more lightweight and efficient 3DGS rendering (Fan et al., 2024;

Liu et al., 2025; Feng et al., 2025a; Gao et al., 2025a)), expand to broader domains and longer video contexts, and build stronger benchmarks for holistic 3D spatial understanding.

This work uses publicly available data sources and does not require collecting new sensitive personal data. Nonetheless, similar technology could be misused for privacy-invasive reconstruction of personal spaces; we encourage responsible deployment with consent, data governance, and appropriate safeguards.

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

# A. Appendix

## A.1. Why 3D Multi-view Merges

Although the masks generated by SAM3 provide instance-level attributes at the image level, occlusions often cause SAM3 to fragment the same object into multiple instances—for example, in Section A.1, a complete bed is detected as two separate beds. Thus, directly adopting SAM3's instance predictions is unreliable; robust 3D clustering and fusion remain essential.

For effective 3D clustering and merging, high geometric accuracy—namely, precise depth estimations—is crucial. As demonstrated in Figure 11, without the multi-view geometric constraint, estimated depth maps suffer from ghosting and aliasing artifacts, causing different objects to become spatially entangled and ultimately mis-merged as a single instance.

Furthermore, as shown in Figure 14, we provide more qualitative examples of 3D grounding box predictions across diverse object categories, underscoring the robustness and generalization capability of our method.

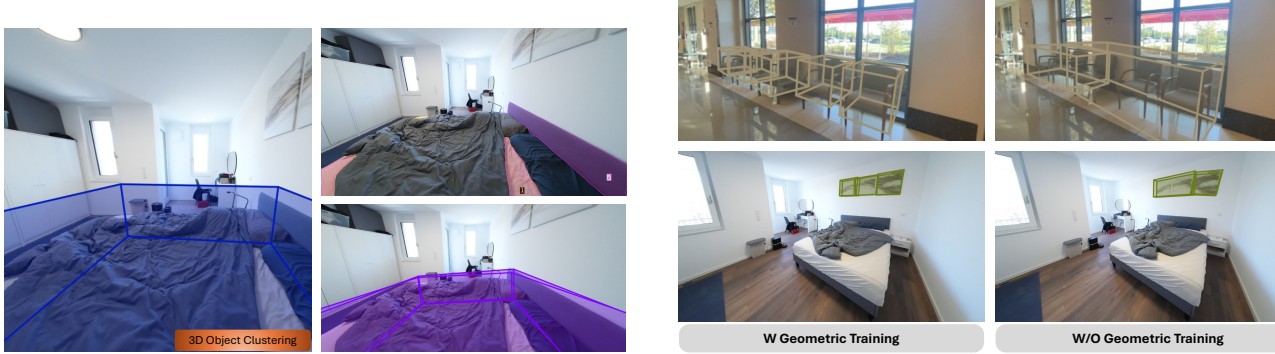

*Figure 11.* **Ablation study on 3D multi-view merging and depth refinement.** (a) 3D geometric clustering corrects the fragmentation caused by occlusions in SAM3 image-level instance predictions, merging separated segments into unified object instances. (b) Using GS-refined depth constraints enables more accurate multi-view lifting, which avoids false merges and preserves the integrity of spatial object instances.

## A.2. QA Examples

We present examples of each of our curated QA question type in Figure 15, including: camera rotation, camera movement direction, camera movement distance, camera-object direction, camera-object distance, camera-object distance (global frame), object-object distance, object measurement, object-object direction (local frame), object-object direction (global frame) . As shown in Figure 16, training on the QA dataset can greatly enhance many tasks in MindCube (Yin et al., 2025) and MMSI-Bench (Yang et al., 2025b), especially perspective changing and egocentric imagination related questions. Additionally, we use VLM to describe the obejct's appearance, and use it as a way to reference the object in the question. This improves the appearance and grounding ability of models in spatial question answering, as shown in the bottom-left case in Figure 16.

## A.3. Prompt Templates

We provide the prompt templates for using agent verification in VLM-based confidence filtering and refinements in Figure 12, and the prompt templates for object grounding in 2D image in Figure 13.

## A.4. Scene-Level Refinement Algorithm

We provide the scene-level refinement algorithm in Algorithm 1.

Prompt Template for Agent Verify

SYSTEM:
You are a visual label verifier for a single masked object.
USER:
"You will be given three images: (1) the original image, (2) an overlay image where the target object is highlighted with a colored mask, and (3) a zoomed-in pair showing the original object and the highlighted object.\n"
f'The user-provided label/category to verify is: "{category}".\n\n'
"Task:\n"
"Decide whether the given label correctly describes the object indicated by the HIGHLIGHTED region.\n"
"If it is correct, ACCEPT. If it is incorrect (wrong category), REJECT and provide the correct label you believe fits best.\n\n"
"Guidelines (consistency-and-rewrite mindset):\n"
"1) Focus ONLY on the object covered by the HIGHLIGHTED region. Ignore other objects outside the mask.\n"
"2) Identify the core visible noun/object class of the masked object (the best label).\n"
"3) Treat extra modifiers in the label (color/material/size/position/relations) as constraints ONLY if they are clearly visible.\n"
" If modifiers are wrong or unverifiable but the core object class is correct, still ACCEPT.\n"
"4) If the mask covers only a part of an object but the object class is still clear, judge by the most likely full object.\n"
"5) If the masked region is ambiguous, too small, or does not correspond to a recognizable object, REJECT and set predicted_label to \"unknown\".\n"
"6) Use common-sense synonyms/hypernyms: accept reasonable equivalents (e.g., \"sofa\" vs \"couch\").\n"
" If the label is too specific and not verifiable (e.g., exact brand/model/species), prefer a more general correct label.\n\n"
"Output format (VERY IMPORTANT):\n"
"Return ONLY a JSON object on a single line with these keys:\n"
" - decision: \"ACCEPT\" or \"REJECT\"\n"
" - predicted_label: a short English noun phrase for the masked object class (e.g., \"chair\", \"person\", \"car\").\n"
"Rules:\n"
" - If decision is ACCEPT, predicted_label should be exactly the provided label (category) or its closest normalized form.\n"
" - If decision is REJECT, predicted_label must be your best guess of the correct label, or \"unknown\" if unclear.\n"
"Do not output any extra text after the JSON object. The JSON should be the final part of your response."

*Figure 12.* **Prompt template used for Agent verification.**

---

**Prompt Template for Object Grounding**

SYSTEM:
You are a helpful assistant.
USER:
"You are labeling objects in a video frame. Identify ALL visible objects in the image. "
"Use concise, lowercase English nouns as object class names. Be instance-aware when deciding what is present, "
"but output only class names (no instance IDs) and list each class at most once.\n\n"
"Granularity rules (avoid over-fragmentation):\n"
"- Use whole-object categories at a practical level (e.g., sofa, pillow, bed, shoe, broom).\n"
"- Do NOT split a single object into parts or components (e.g., do not use 'broom handle', 'broom head'; use 'broom').\n"
"- Prefer common household object categories; avoid overly fine-grained subtypes.\n\n"
"No skipping rule:\n"
"- You MUST label every distinct object you can see.\n"
"- If you are unsure of an exact subtype, choose the closest reasonable, still-concrete supercategory (e.g., 'chair' instead of a specific chair type; "
"'bottle' instead of a specific brand; 'bag' instead of an uncertain bag type). Do not omit objects.\n"
"- Do NOT use non-informative words like 'object', 'thing', 'stuff', 'unknown'.\n\n"
"Consistency with previous frames (canonical vocabulary):\n"
f"- Existing labels from previous frames: {previous_text}\n"
"- If any object in the current image matches a previous label in meaning (including synonyms, singular/plural, or minor wording differences), "
"you MUST reuse the previous label exactly and MUST NOT introduce a new synonym.\n"
"- Only create a new label if no existing label can reasonably cover the same object meaning.\n\n"
"Output format:\n"
"- Return a single line of lowercase, comma-separated class names only.\n"
"- Include classes that appeared in previous frames if they also appear here.\n"
"- No explanations, no extra text."

*Figure 13.* **Prompt template used for object grounding in 2D image.**

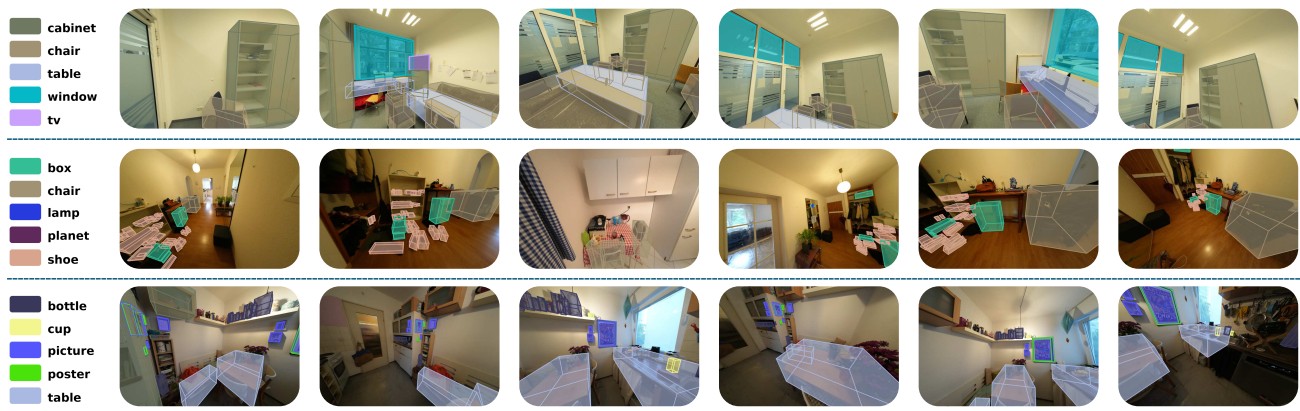

*Figure 14.* **Additional 3D detection visualizations.** We show Holi-Spatial predictions across diverse indoor scenes, demonstrating multi-view consistent localization and robust detection.

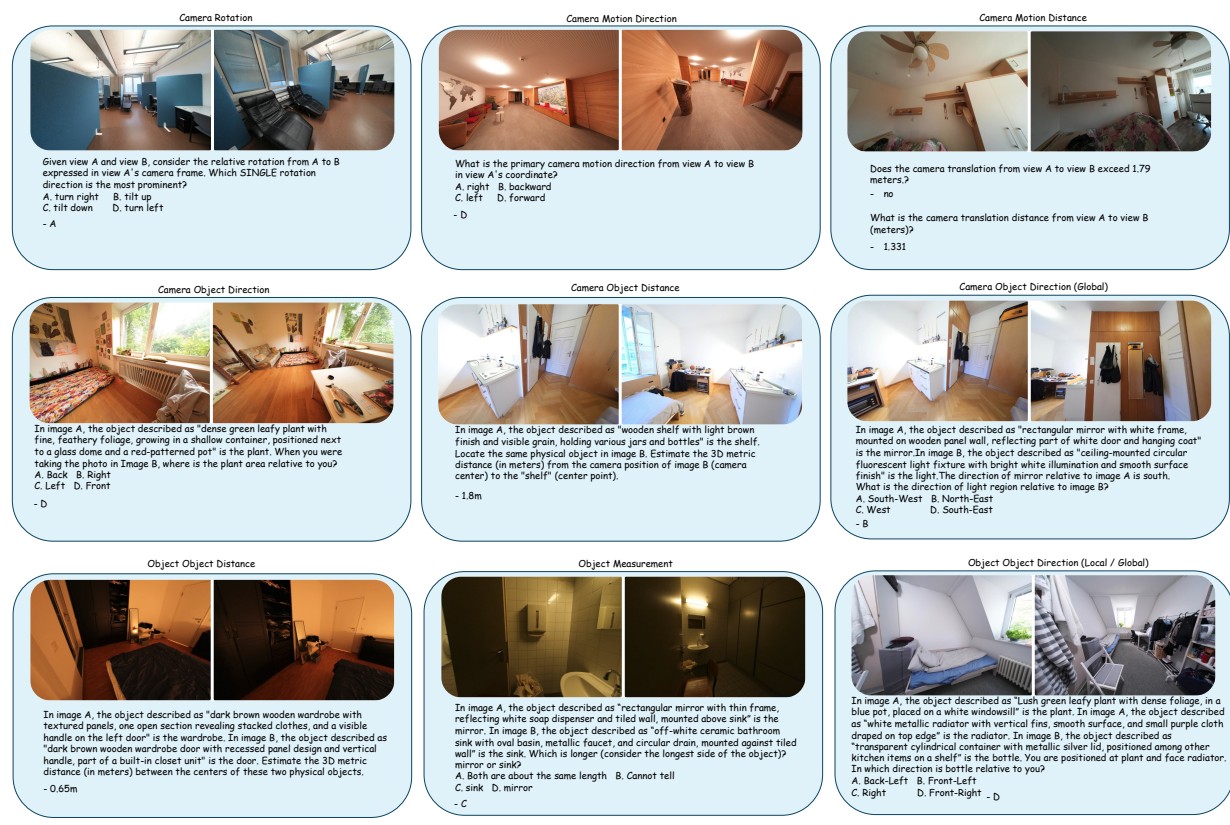

*Figure 15.* **Examples of 10 types of curated spatial QA pairs in Holi-Spatial.**

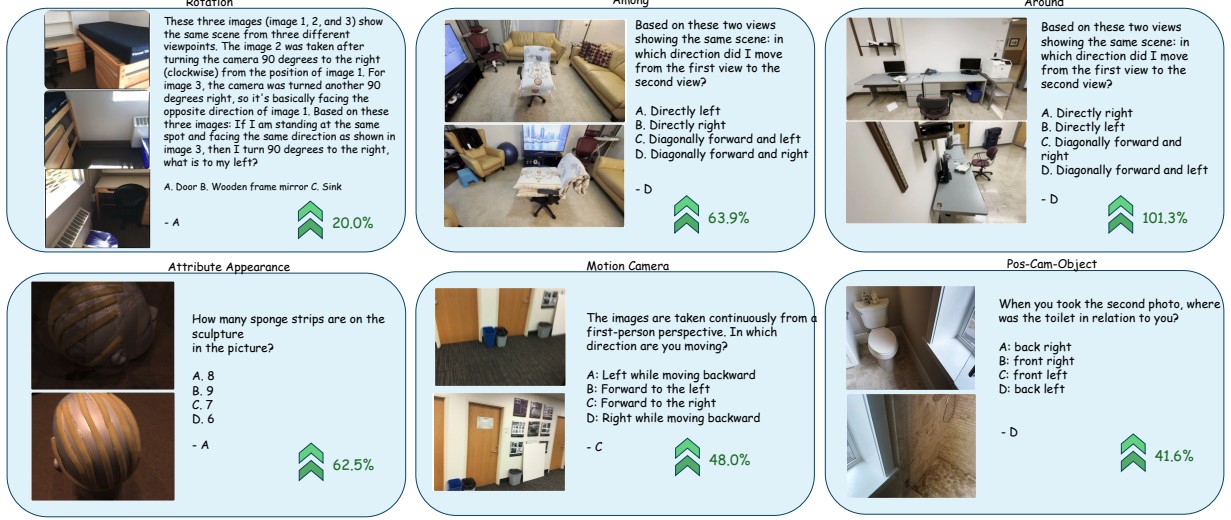

*Figure 16.* **Training with our curated QA data improves the three types of spatial reasoning tasks in MindCube (Yin et al., 2025) (upper row) as well as in MMSI-Bench (Yang et al., 2025b) (lower row).**

---

**Algorithm 1** Scene-level Refinement via 3D Merging and VLM-based Verification

---

1: **Input:** keyframes $\{I_t\}_{t=1}^T$, camera parameters $\{\Pi_t\}_{t=1}^T$, refined depth maps $\{D_t\}_{t=1}^T$, per-frame label sets $\{\mathcal{L}_t\}_{t=1}^T$.
2: **Input:** $\mathrm{SAM3}(I_t, \ell) \rightarrow \{(m_{t,\ell,k}, s_{t,\ell,k})\}_{k=1}^{K_{t,\ell}}$, where $m$ is a 2D mask and $s \in [0,1]$ is confidence.
3: **Input:** thresholds $\tau_{\mathrm{iou}}$, $\tau_{\mathrm{low}} < \tau_{\mathrm{high}}$.
4: **Output:** validated instance set $\mathcal{O}$; each item $(\ell, B_g, t^\star, m^\star)$.
5: **(1) Lift 2D instances to 3D candidates (per label)**
6: Initialize candidate pools $\mathcal{C}_\ell \leftarrow \emptyset$ for all labels $\ell$.
7: **for** $t = 1$ **to** $T$ **do**
8:     **for all** $\ell \in \mathcal{L}_t$ **do**
9:         $\{(m_{t,\ell,k}, s_{t,\ell,k})\}_{k=1}^{K_{t,\ell}} \leftarrow \mathrm{SAM3}(I_t, \ell)$
10:         **for** $k = 1$ **to** $K_{t,\ell}$ **do**
11:             $P_{t,\ell,k} \leftarrow \mathrm{BackProject}(m_{t,\ell,k}, D_t, \Pi_t)$
12:             $B_{t,\ell,k} \leftarrow \mathrm{BBox}(P_{t,\ell,k})$
13:             $c_{t,\ell,k} \leftarrow (P_{t,\ell,k}, B_{t,\ell,k}, s_{t,\ell,k}, t, m_{t,\ell,k})$
14:             $\mathcal{C}_\ell \leftarrow \mathcal{C}_\ell \cup \{c_{t,\ell,k}\}$
15:         **end for**
16:     **end for**
17: **end for**
18: **(2) Multi-view merging within each label (3D IoU)**
19: Initialize merged groups $\mathcal{G} \leftarrow \emptyset$.
20: **for all** label $\ell$ such that $\mathcal{C}_\ell \neq \emptyset$ **do**
21:     $\mathcal{G}_\ell \leftarrow \mathrm{MergeByIoU3D}(\mathcal{C}_\ell, \tau_{\mathrm{iou}})$
22:     **for all** group $g \in \mathcal{G}_\ell$ **do**
23:         $\mathcal{G} \leftarrow \mathcal{G} \cup \{(\ell, g)\}$                           *// each g is a set of candidates $c_{t,\ell,k}$*
24:     **end for**
25: **end for**
26: **(3) Confidence gating and VLM-based verification**
27: Initialize $\mathcal{O} \leftarrow \emptyset$.
28: **for all** instance group $(\ell, g) \in \mathcal{G}$ **do**
29:     $c^\star \leftarrow \arg\max_{c \in g} s(c)$                                 *// canonical view*
30:     $(P^\star, B^\star, s^\star, t^\star, m^\star) \leftarrow c^\star$
31:     $P_g \leftarrow \bigcup_{c \in g} P(c)$
32:     $B_g \leftarrow \mathrm{BBox}(P_g)$
33:     **if** $s^\star \geq \tau_{\mathrm{high}}$ **then**
34:         $\mathcal{O} \leftarrow \mathcal{O} \cup \{(\ell, B_g, t^\star, m^\star)\}$
35:     **else if** $s^\star < \tau_{\mathrm{low}}$ **then**
36:         **continue**                                    *// discard*
37:     **else**
38:         $y \leftarrow \mathrm{VLMVerify}(I_{t^\star}, m^\star, \ell)$
39:         **if** $y = \texttt{true}$ **then**
40:             $\mathcal{O} \leftarrow \mathcal{O} \cup \{(\ell, B_g, t^\star, m^\star)\}$
41:         **end if**
42:     **end if**
43: **end for**
44: **return** $\mathcal{O}$.

---

