# OpenReview forum: "Holi-Spatial: Evolving Video Streams into Holistic 3D Spatial Intelligence"
_ICML.cc/2026/Conference — ICML 2026 spotlight_

### Official Review · Reviewer_au8D · 2026-03-08

**Soundness:** 3
**Presentation:** 3
**Significance:** 3
**Originality:** 3
**Overall Recommendation:** 5
**Confidence:** 4

**Summary:**

This paper presents Holi-Spatial, a fully automated pipeline that converts raw video streams into 3D geometry and multi-level spatial annotations, and uses this pipeline to build the Holi-Spatial-4M dataset. The method consists of three stages. It first performs geometric optimization using SfM, monocular depth priors, and 3DGS refinement to obtain more consistent geometry and depth. It then samples keyframes and applies a VLM together with SAM3 to produce open-vocabulary 2D masks, which are lifted into 3D. Finally, it refines scene annotations through multi-view merging, confidence-based triage, and VLM-agent verification, producing 3D boxes, captions, 3D grounding instances, and spatial QA pairs. The paper further evaluates annotation quality on ScanNet, ScanNet++, and DL3DV-10K, and fine-tunes Qwen3-VL on the resulting data for spatial reasoning and 3D grounding tasks.

**Compliance With Llm Reviewing Policy:**

Affirmed.

**Final Justification:**

This paper is a technically solid and useful contribution to scalable 3D spatial data construction, and the authors’ rebuttal addressed my main concerns sufficiently to improve my confidence in the paper’s positioning, soundness, and practical significance, leading me to a more positive final recommendation.

**Key Questions For Authors:**

1. Could the authors briefly clarify what they see as the main bottleneck in current spatial intelligence datasets, and why existing datasets or annotation pipelines are insufficient to address it?

2. Since the goal is to scale 3D spatial annotation, why is the dataset construction based on ScanNet, ScanNet++, and DL3DV-10K, which already provide 3D or RGB-D scene data? Could the authors justify this choice and discuss whether the pipeline can also be applied to datasets without existing 3D annotations?

3. Could the authors analyze possible data leakage, pretraining overlap, or dataset-specific priors more carefully, and explain why these factors do not materially affect the conclusions?

4. Could the authors explain why the gain on MMSI-Bench is relatively small compared with the stronger gains on other benchmarks, and what this implies about the actual benefit of the curated dataset?

Convincing answers to these questions could lead me to revise my overall score upward.

**Limitations:**

Yes

**Strengths And Weaknesses:**

Strengths

S1: The paper addresses an important bottleneck in spatial intelligence, namely the lack of scalable, fine-grained 3D spatial supervision. A fully automated pipeline that expands raw video streams into richer 3D geometry, object-level annotations, grounding pairs, and spatial QA is a meaningful direction and could be useful for future data-centric research in this area.

S2: Rather than introducing only an isolated module, the paper presents a full pipeline covering geometric reconstruction, keyframe-level perception, 2D-to-3D lifting, multi-view merging, confidence-based refinement, and annotation synthesis. For a dataset / pipeline paper, this level of completeness is a notable strength, and the overall methodology is reasonably easy to follow.

S3: The paper reports 12K optimized 3DGS scenes and more than 4M annotations across multiple supervision types. It also evaluates both annotation quality and downstream VLM fine-tuning performance, which makes the experimental story more complete than a purely descriptive dataset release.

Weakness

W1: The work is clearly hard-working and well engineered, but the novelty is a littile bit limited. The pipeline is mostly built from existing components and does not introduce a clearly new core method. At the same time, the paper does not contribute fundamentally new data, since it mainly re-annotates existing public datasets. As a result, neither the methodological contribution nor the dataset contribution feels strong enough on its own.

W2: The data source does not fully match the paper’s claim. The paper strongly emphasizes raw videos and large-scale 3D spatial annotation, but Holi-Spatial-4M is built from ScanNet, ScanNet++, and DL3DV-10K, which are already existing 3D or RGB-D datasets. This weakens the claim because these datasets already come with useful geometry and annotations. As a result, the work feels more like annotation expansion on existing 3D datasets than a truly new large-scale 3D data construction pipeline from open-world raw videos.

W3. I am concerned about possible data leakage or strong prior exposure. The pipeline relies on strong upstream models such as DA3, Gemini, SAM3, and Qwen3-VL, while both construction and evaluation are done on very common public datasets like ScanNet and ScanNet++. The paper does not clearly explain how it rules out pretraining overlap or strong dataset-specific priors, so the results are harder to fully trust.

Minor:

Please analyze why the gain on MMSI-Bench is relatively small compared with the stronger gains on other benchmarks. Without such analysis, it is hard to understand what insight this experiment is meant to provide.

---

> ### Author Rebuttal · Authors · 2026-03-31
>
> **Q1**: Could the authors briefly clarify what they see as the main bottleneck in current spatial intelligence datasets, and why existing datasets or annotation pipelines are insufficient to address it?
>
> **A1**:
> The primary bottleneck in current spatial intelligence datasets lies in their **scalability and the accuracy of geometric relationships**. Manual annotation of large-scale datasets is prohibitively expensive and time-consuming, making it difficult to scale up. Furthermore, accurate geometry is fundamental for true spatial understanding. Spatial intelligence relies on precise, unambiguous geometric relationships, but current datasets often have errors or ambiguities in this regard. As most existing datasets are based on manual labeling, they inherently face a scalability barrier. Additionally, there is a lack of reliable methods for lifting widespread 2D data to accurate 3D geometry, which further limits progress in spatial intelligence research.
>
> **Q2**: Since the goal is to scale 3D spatial annotation, why is the dataset construction based on ScanNet, ScanNet++, and DL3DV-10K, which already provide 3D or RGB-D scene data? Could the authors justify this choice and discuss whether the pipeline can also be applied to datasets without existing 3D annotations?
>
> **A2**:**We did not use the existing depth or point cloud data from these datasets in any of our initialization or supervision processes.** The reason we utilize them is that their ground-truth point clouds or depth data provide a better benchmark for evaluating our results. Besides, our annotations have more details than the original ones. For example, in the ScanNet++ dataset, in scene 0a5c013435, there are 27 objects annotated by human annotators, while our approach identifies and annotates 49 objects, discovering additional categories such as 'mop, cart, bottle' that were missing from the manual labels.  Moreover, the depth generated by our method is more complete and more detailed than that provided by the original dataset.  The corresponding visualizations are available at the anonymous link: https://anonymous.4open.science/w/review-4C20/
>
>
> **Q3**: Could the authors analyze possible data leakage, pretraining overlap, or dataset-specific priors more carefully, and explain why these factors do not materially affect the conclusions?
>
> **A3**:
> Data Leakage Clarification:
> - Reconstruction Perspective: According to the official DA3 paper, the model was specifically trained on ScanNet++ rather than the ScanNet. Therefore, there is no data leakage or overlap affecting the results on the ScanNet benchmark.
> - Perception Perspective: SAM3 is pre-trained on large-scale 2D image-mask datasets such as SA-1B and COCO. In contrast, our evaluations are conducted on 3D datasets, including ScanNet, ScanNet++, and DL3DV. It represents a fundamentally different distribution from SAM’s 2D training sets. This cross-domain evaluation ensures there is no risk of data contamination or leakage.
>
> Pertraining Overlap and  Dataset-Specific Prior Clarification:
> - Curated Data vs. Test Benchmarks: Our Holispatial-4M dataset is primarily constructed from ScanNet, ScanNet++, and DL3DV. In contrast, we evaluate our fine-tuned VLM on MMSI-Bench and MindCube, which are built upon entirely different sources, including Matterport3D, Waymo, AgiBot-World, and Ego4D. These benchmarks encompass vastly different domains, ranging from indoor navigation, autonomous driving,  robotics, and egocentric domains. The significant domain gap between our training data and these evaluation suites ensures that our results are not the product of data overlap, but rather a reflection of the model's robust cross-domain generalization.
>
>
> **Q4**:  MMSI-Bench is relatively small.
>
> **A4**: MMSI-Bench is a rigorously human-curated dataset characterized by complex spatial reasoning challenges. While the performance of most open-source models is clustered within the 28.0%–31.0% accuracy range, our method achieves an accuracy of 32.6%. This result is on par with the state-of-the-art SenseNova-SI-1.1-Qwen2.5-VL-7B with the intensive, task-specific QA fine-tuning process. Furthermore, MMSI-Bench includes several motion-related tasks for which our method was not originally designed, which naturally imposes certain limitations on the performance of our fine-tuned VLM.
> Moreover, we additionally evaluate 2 more spatial benchmarks, i.e., ViewSpatial and SparBench-tiny, which consistently demonstrate the significant performance gains provided by our data.
>
> | Model | ViewSpatial | SparBench-tiny |
> |---|---:|---:|
> | VST-SFT-3B | 52.89 | 37.71 |
> | Cambrian-S-3B | 40.97 | 33.02 |
> | VST-SFT-7B | 50.53 | 46.61 |
> | Cambrian-S-7B | 41.28 | 37.91 |
> | Intern3-VL-8B | 38.66 | 35.86 |
> | Spatial-MLLM | 34.66 | 35.31 |
> | Qwen3-VL-8B | 42.35 | 39.83 |
> | Qwen3-VL-8B + Ours | **50.12** | **46.28** |

---

> > ### Author Rebuttal · Reviewer_au8D · 2026-04-01
> >
> > The authors have fully addressed my concerns in the rebuttal, and I believe this work should be accepted, so I am increasing my score accordingly.

---

> > > ### Author Response · Authors · 2026-04-04
> > >
> > > We greatly appreciate your constructive feedback. Based on your suggestions, we will incorporate the corresponding experiments and explanation into the main paper and supplementary material. Please feel free to let us know if there are any further questions or suggestions. Thank you again for your valuable time and suggestions.

---

### Official Review · Reviewer_8e5b · 2026-03-12

**Soundness:** 4
**Presentation:** 3
**Significance:** 3
**Originality:** 2
**Overall Recommendation:** 5
**Confidence:** 4

**Summary:**

This paper presents Holi-Spatial, a fully automated pipeline that converts raw video streams into holistic 3D spatial annotations without human intervention. The pipeline combines per-scene 3DGS-based geometric optimization, image-level open-vocabulary perception using Gemini3-Pro and SAM3, and scene-level merging, filtering, verification, and caption generation with VLM-based modules, ultimately producing multi-level supervision including rendered depth, 2D masks, 3D bounding boxes, grounding annotations, and spatial QA pairs. Based on this pipeline, the authors construct Holi-Spatial-4M.

**Compliance With Llm Reviewing Policy:**

Affirmed.

**Final Justification:**

I support accepting this solid paper.

**Key Questions For Authors:**

- Clarify the exact model used for the Stage-3 verification/refinement agent. Is it the same as Gemini3-Pro in Stage 2, or a different VLM?
- Since the pipeline starts from videos, do the authors plan to extend Holi-Spatial toward dynamic / 4D supervision, such as object trajectories, interactions, or temporal state changes, rather than mainly scene-level static annotations?

**Limitations:**

Yes

**Strengths And Weaknesses:**

### Strengths
- The paper tackles an important bottleneck in spatial intelligence: the scarcity and imbalance of raw spatial data, as well as the limited semantic coverage of existing curated 3D datasets. This is a meaningful for the community.
- The proposed framework is broad and genuinely unified. Holi-Spatial supports a wide range, including depth, 2D segmentation, 3D detection, grounding, and QA.
- The dataset scale and richness are impressive.

### Major Weaknesses
- The scalability story is promising but not yet fully convincing. A core part of the pipeline still depends on per-scene 3DGS optimization, and the paper itself notes that optimization-driven 3DGS is time-consuming. This may block the further scaling of the datasets.
- The scene-level refinement stage appears effective, but methodologically it is still somewhat heuristic, relying on IoU-based merging, confidence thresholds, and a fallback VLM verification step for borderline cases. The ablation shows that confidence filtering improves precision but hurts recall, and the agent-based recovery is then introduced to compensate. This engineering design is practical, but it is less satisfying as a clean, principled formulation.
- The object-grounding prompt explicitly asks the model to label "every distinct object you can see" and forbids outputs such as “unknown,” “thing,” or “stuff.” For ambiguous, tiny, or occluded regions, this may encourage over-labeling or hallucinated categories instead of calibrated abstention.

### Minor Weaknesses
- Although the input source is video, the resulting supervision is still centered on scene-level 3D reconstruction and static object-centric annotations, rather than explicit modeling of dynamic objects or true 4D world evolution. The authors also acknowledge degradation on dynamic objects and list longer video contexts as future work.
- The paper should clarify whether the Stage-3 VLM verifier is the same model as the Stage-2 VLM (Gemini3-Pro) or a different one.
- It would also be helpful to show some failure cases as in the limitation parts: highly ambiguous, occluded, dynamic, or low-resolution objects.

---

> ### Author Rebuttal · Authors · 2026-03-31
>
> **Q1**: Scalability concern.
>
> **A1**:
> The 3DGS optimization part is indeed the main time-consuming step in our pipeline. To mitigate this, we apply several engineering strategies to reduce the overall time cost:
> -During training, we perform frame sampling on videos (for example, on the ScanNetv2 dataset, we sample one frame every 15 frames). This reduces the number of images needed for training each scene to below 1000 or even fewer.
> - We further compress training time through multi-GPU parallelization; with 16 H800 GPUs, over 500 scenes can be processed within 24 hours.
> - Additionally, as shown in Table 5 by the comparison between ID1 and ID2, 3DGS training is clearly necessary for optimal performance in our current pipeline.
> - Furthermore, 3DGS serves as a highly valuable representation with significant utility in tasks such as simulator and world modeling. Compared to the reconstructed 3DGS provided by ScanNet++ and the recent SceneSplat / SceneSplat++ datasets, our results demonstrate superior geometric accuracy and rendering fidelity.
> Notably, recently emerged datasets such as Xperience-10M [1], EgoScale [2], and EgoVerseAI [3] utilize advanced hardware to provide high-fidelity depth and pose metadata. Our method seamlessly integrates with these platforms, bypassing the computationally expensive 3DGS reconstruction pipeline typically required.
>
> [1] https://huggingface.co/datasets/ropedia-ai/xperience-10m
>
> [2] https://research.nvidia.com/labs/gear/egoscale
>
> [3] https://egoverse.ai
>
>
> **Q2**: Method seems heuristic rather than principled.
>
> **A2**:: Our main objective is to design a practical and feasible solution and to demonstrate, through experiments, the strong potential and effectiveness of current models as alternatives to manual dataset construction. At this stage, our goal is not to impose a fully unified formulation at the expense of robustness, but rather to develop a reliable decision pipeline. The refinement stage follows a coarse-to-fine principle: confident predictions are accepted directly, cross-view overlaps are consolidated through scene-level merging, and only uncertain cases are escalated to an additional verifier. This is an intentional cascade design for uncertainty management, as opposed to a collection of isolated engineering tricks.
>
> **Q3**: Grounding prompt may encourage over-labeling instead of calibrated abstention.
>
> **A3**: Thank you for your suggestion. As you pointed out, VLM may indeed produce some hallucinations, which we also observed in our experiments. However, this does not affect our overall robustness. For hallucinated labels, the SAM3 segmentation does not produce masks or only generates masks with low confidence. Together with the later-stage VLM recall mechanism, these cases are handled effectively. In the first grounding stage, our approach is to encourage the model to identify as many instances as possible, even if some are incorrect, as it does not undermine the robustness of our pipeline.
>
>
> **Q4**: Explicit modeling of dynamic objects or true 4D world evolution.
>
> **A4**: While our current approach is specifically designed for static scenes, we have included a dynamic scenario demo on our anonymous rebuttal website [https://anonymous.4open.science/w/review-4C20/]. It demonstrates that our method effectively handles static environments. It also reveals a surprising generalization capability for dynamic objects. Although it effectively tracks movement in many cases, it is currently limited by occasional misidentifications and drift over long-term sequences, which we aim to address in future work.
>
> **Q5**: Details on Stage 3.
>
> **A5**:  In Stage 3, the model we use is QwenVL-3-32B-Thinking.
>
> **Q6**: Show failure cases: highly ambiguous, occluded, dynamic, or low-resolution objects.
>
> **A6**: We provide several failure cases in the anonymous link https://anonymous.4open.science/w/review-4C20/, including examples under dynamic objects and low-resolution settings. In the low-resolution case, the blueberry object is not effectively recognized due to the limited visual detail. We also observe that the recognition of dynamic objects in videos is not sufficiently temporally consistent. These limitations reflect the current weaknesses of our method and point to important directions for future improvement.

---

> > ### Author Rebuttal · Reviewer_8e5b · 2026-04-01
> >
> > Thanks for the authors' explanation. It directly addresses my concerns, especially the dynamic cases and the scalability concerns. I support accepting this solid paper.

---

> > > ### Author Response · Authors · 2026-04-04
> > >
> > > We greatly appreciate your constructive feedback. Based on your suggestions, we will incorporate the corresponding experiments and explanation into the main paper and supplementary material. Please feel free to let us know if there are any further questions or suggestions. Thank you again for your valuable time and suggestions.

---

### Official Review · Reviewer_jxNP · 2026-03-13

**Soundness:** 4
**Presentation:** 4
**Significance:** 4
**Originality:** 3
**Overall Recommendation:** 5
**Confidence:** 4

**Summary:**

In this work, the authors propose Holi-Spatial, the first fully automated pipeline that converts raw video streams into holistic 3D spatial annotations without human intervention. The method demonstrates exceptional performance in data curation quality. Using this pipeline, the authors collect a large-scale, high-quality 3D semantic dataset, Holi-Spatial-4M. VLMs achieve substantial improvements in spatial reasoning capabilities after training on Holi-Spatial-4M.

**Compliance With Llm Reviewing Policy:**

Affirmed.

**Key Questions For Authors:**

How is the open-vocabulary GT annotated (L113-L115)? Could the authors provide more details? Re-annotating datasets is usually very costly.

**Limitations:**

yes

**Strengths And Weaknesses:**

Strengths:
1. Holi-spatial is the first fully automated pipeline that converts raw video streams into 3d annotations without any human-in-the-loop design. This is remarkable, considering the excellent data quality demonstrated by quantitative and qualitative results.
2. The pipeline is scalable since it’s fully automated. This makes the pipeline highly significant for the 3D vision-language field.
3. The pipeline is easy to understand and well-motivated. Holistic ablation studies demonstrate the effectiveness of the pipeline design.
4. The paper introduces a large-scale 3D semantic dataset, Holi-Spatial-4M, which is a valuable research resource for the 3D research community.

Weaknesses:
1. As the authors mention, the pipeline can be computationally expensive. Besides, the usage of VLMs may also be costly. Authors could quantitatively measure these costs to provide more information about the pipeline’s time & financial efficiency.
2. L325-L326: In the second line of Figure 6, it seems that SAM3 successfully segments the distant mirror, just as the proposed pipeline Holi-Spatial. There might be a mistake here.

---

> ### Author Rebuttal · Authors · 2026-03-31
>
> **Q1**: Quantitatively measure.
>
> **A1**: Based on analysis across our training scenes, we count that each scene in Step 1 requires 247,668 tokens for processing (Financial cost excluding tokens used in Stage 3, for which we employ open-source models and cost 141,313 tokens). In terms of time efficiency, end-to-end processing of a single scene takes about 2 hours. Our system is optimized for parallelism: we allocate 8 GPUs for deploying open-source models to facilitate pipeline calls, while the remaining 24 GPUs are used for parallel end-to-end processing. This setup allows us to process around 500 scenes per day under typical conditions.
>
>
> | Metric                      | Step 1  | Step 2 | Step 3  |
> |-----------------------------|---------|--------|---------|
> | Token Count                 | 247,668 | 0      | 141,313 |
> | End-to-End Processing Time  | 42 min  | 2 min  | 15 min  |
>
> **Q2**: Details of open-vocabulary GT annotated.
>
> **A2**: We first use Gemini-3-Pro to caption all objects in the images, followed by manual verification to remove erroneous or add missing instance categories per image. We then use the Roboflow labeling platform (with SAM 3) to segment 2D masks for each object in every image. For objects that the model fails to detect, we enforce segmentation using point prompts. For 3D bounding boxes, we manually annotate the ground truth using the LabelCloud platform, with object positions and categories guided by the 2D instance masks obtained above.
>
> **Q3**: Inaccurate description in Figure 6
>
> **A3**: In Figure 6,  it's our inaccurate description: "the distant window in the third row" was mistakenly written as "mirror". We will fix the inaccurate description in our Camera Ready version.

---

> > ### Author Rebuttal · Reviewer_jxNP · 2026-04-03
> >
> > I thank the authors for their detailed rebuttal. The quantitative analysis of computational costs (token usage and processing time per stage) and the clear explanation of the open-vocabulary GT annotation process have fully addressed my questions. The acknowledgment of the Figure 6 description error is also appreciated. My concerns are resolved.

---

> > > ### Author Response · Authors · 2026-04-04
> > >
> > > We greatly appreciate your constructive feedback. Based on your suggestions, we will incorporate the corresponding experiments and details into the main paper and supplementary material. Please feel free to let us know if there are any further questions or suggestions. Thank you again for your valuable time and suggestions.

---

### Official Review · Reviewer_4Wvi · 2026-03-17

**Soundness:** 3
**Presentation:** 4
**Significance:** 3
**Originality:** 3
**Overall Recommendation:** 5
**Confidence:** 3

**Summary:**

The authors propose a fully automated framework to scale up spatial data and then evaluate it's efficacy by training VLMs with the data and evaluating it on different 3D and 2D downstream tasks. The motivation behind this proposal is that raw spatial data is scarce, imbalanced, and lack semantic annotations. The bigger reason behind why it's scarce is because previous methods have had to rely on expensive methods such as collecting data from 3D sensors and using human annotation. By relying on existing excellent AI tools, the authors propose a combination of those tools to come up with a data pipeline that can work with video data (which is widely available) and curates and annotates the collected video data. By using the data from the pipeline, the resulting model outperforms baselines across tasks such as depth estimated, spatial reasoning, 3D object detection, 3D grounding and 2D object segmentation.

**Compliance With Llm Reviewing Policy:**

Affirmed.

**Key Questions For Authors:**

- Please explain why you could only conduct the Framework Evaluation on 10 samples? Could you provide error bars so we can see the variance in the results?
- Can you clarify how you came to the threshold values for multi-view instance merging or confidence-based filtering?

**Limitations:**

Yes.

**Strengths And Weaknesses:**

Soundness:
The methods introduced are well-motivated and explained clearly. There are also ablations explaining why each method is necessary to the final downstream performance of the fine-tuned model. One small missing detail is the justification for the thresholds used in the filtering mechanisms (confidence-based filtering or multi-view instance merging). However, for the Framework Evaluation, 10 scenes being used to evaluate might be too small for the results to be convincing. Potentially evaluating on 50 scenes might be more convincing. Otherwise, the evaluation is thorough, involving comparison to many other baselines and on 3 different tasks. The VLM Finetuning Evaluation is designed well by evaluating on both 3D grounding and spatial reasoning and is compared to many other baselines.

Presentation:
The submission was written with many well-justified details, sections were clearly explained and the paper was well structured.

Significance:
The paper does address a relevant problem of the resource-taxing process of collecting high-quality raw spatial data with annotations. The authors propose a framework to collect, curate and annotate spatial data without human-in-the-loop or using 3D sensors, both of which can be expensive. Moreover, the downstream performance on different tasks show improvement from existing baselines after having finetuned Qwen models on the data.

Originality:
The paper introduces a novel method to scale up annotated spatial data with a fully automated scalable method that doesn't require expensive human annotations or 3D sensor data. In this method, it involves highlighting important properties of existing adjacent methods that can be combined to make large-scale high quality spatial data more accessible.

---

> ### Author Rebuttal · Authors · 2026-03-31
>
> **Q1**: threshold of confidence-based filtering and multi-view instance merging.
>
> **A2**: These two values were determined empirically based on our experimental observations.
>
> | Confidence Filter Threshold | AP25  | AP50  |
> |-------|-------|-------|
> | 0.7                        | 76.15 | 68.94 |
> | 0.8                        | 79.13 | 69.56 |
> | 0.9                        | 81.06 | 70.05 |
> | 0.95                       | 79.84 | 69.02 |
>
>
> We conducted ablation experiments with different confidence filter thresholds on ScanNet++. As shown, setting the filter to 0.9 achieves the highest AP when combined with the subsequent VLM recall mechanism.
>
>
> | Merge IOU Threshold | AP25  | AP50  |
> |-------|-------|-------|
> | 0.1                        | 77.18 | 68.45 |
> | 0.2                        | 81.06 | 70.05 |
> | 0.3                        | 77.84 | 69.24 |
>
> As shown in the table, setting the merge IOU threshold to 0.2 yields the highest accuracy, indicating that this value best balances merging true duplicates without merging distinct instances.
>
>
> **Q2**:  10 sample evaluations and error bars to see the variance in the results.
>
> **A2**: Producing high-quality, fine-grained ground-truth annotations is highly labor-intensive. For each scene, we provide both 2D and 3D annotations, including 2D segmentation masks and 3D bounding boxes for every visible object. As a result, creating detailed annotations for a single scene requires over two hours of manual effort.
> However, since we prioritize designing a robust and effective pipeline, we do not invest extensive effort in constructing large-scale ground-truth scenes.
>
> Nevertheless, to further demonstrate the robustness of our method, we also manually annotated **5 more scenes**. The results are consistent with those of the main experiments, confirming the stability and generalizability of our approach.
>
> | Metric | Dataset | 10 scenes | 15 scenes |
> |--------|---------|-----------|-----------|
> | 2D segmentation | ScanNet | 0.79 ± 0.010 | 0.77 ± 0.008 |
> | 3DAP@25 | ScanNet | 76.60 ± 0.41 | 79.29 ± 0.39 |
> | 2D segmentation | ScanNet++ | 0.64 ± 0.012 | 0.66 ± 0.0011 |
> | 3DAP@25 | ScanNet++ | 81.06 ± 0.63 | 80.76 ± 0.57 |
> | 2D segmentation | DL3DV | 0.71 ± 0.017 | 0.72 ± 0.014 |
> | 3DAP@25 | DL3DV | 62.89 ± 1.55 | 60.21 ± 1.21 |

---

> > ### Author Rebuttal · Reviewer_4Wvi · 2026-04-04
> >
> > Thank you for the detailed response. All my questions have been addressed adequately. I'll keep my score!

---

> > > ### Author Response · Authors · 2026-04-04
> > >
> > > We greatly appreciate your constructive feedback. Based on your suggestions, we will incorporate the corresponding clarifications and additions into the main paper and supplementary material. Please feel free to let us know if there are any further questions or suggestions. Thank you again for your valuable time and feedback.

---

### Decision · Program_Chairs · 2026-04-30

**Decision:**

Accept (spotlight)

**Comment:**

All reviewers agreed that it is a well written and solid paper. It makes great contribution for large scale 3D spatial data construction. The authors’ rebuttal is sufficient and clearly, which addresses the reviewers' concerns. Meta agreed to accept the paper!